# `HTRON`: Efficient Outdoor Navigation with Sparse Rewards via Heavy Tailed Adaptive Reinforce Algorithm

**Kasun Weerakoon,**[*] **Souradip Chakraborty**[*]**, Nare Karapetyan, Adarsh Jagan Sathyamoorthy, Amrit Singh Bedi and Dinesh Manocha**.

**Abstract:** We present a novel approach to improve the performance of deep reinforcement learning (DRL) based outdoor robot navigation systems. Most, existing DRL methods are based on carefully designed dense reward functions that learn the efficient behavior in an environment. We circumvent this issue by working only with sparse rewards (which are easy to design), and propose a novel adaptive **H**eavy-**T**ailed **R**einforce algorithm for **O**utdoor **N**avigation called `HTRON`. The key idea in this work is to utilize heavy-tailed policy parametrizations which implicitly induces exploration in sparse reward settings. We evaluate the performance of `HTRON` against Reinforce, PPO and TRPO algorithms in three different outdoor scenarios: goal-reaching, obstacle avoidance, and uneven terrain navigation. We observe in average an increase of $34.41\%$ in terms of success rate, $15.15\%$ decrease in the average time steps taken to reach the goal, and $24.9\%$ decrease in the elevation cost compared to the navigation policies obtained by the other methods. Further, we demonstrate that our algorithm can be transferred directly into a Clearpath Husky robot to perform real-world outdoor terrain navigation.

**Keywords:** Navigation, Deep Reinforcement Learning, Heavy Tailed Policy

## 1 Introduction

Autonomous robot navigation in complex outdoor environments has been an active area of research. The resulting navigation systems are used for different applications, including delivery [1], search and rescue [2], planetary explorations [3], etc. Such applications require robots to gather useful information efficiently from the environment to make intelligent navigation decisions [4]. To this end, Deep Reinforcement Learning (DRL) techniques have been widely employed in recent robotic systems due to their inherent exploration and exploitation capabilities to gather necessary information from the learning environments [5, 6, 7, 8, 9, 10]. Nevertheless, a major challenge for DRL based algorithms is dealing with sparse rewards in continuous state and actions spaces. In particular, the rewards are sparse in navigation settings because it's only available on reaching a goal (positive reward) or hitting an obstacle (negative rewards) [11]. Hence, training DRL policies under sparse reward settings oftentimes lead to instabilities and convergence issues that can significantly degrade consistent performance, especially in navigation applications [12, 13].

To improve the training efficiency and deal with sparse rewards, two popular methods in the literature are reward shaping [14] and demonstration-guided learning [15, 16]. The primary objective of rewards shaping-based methods is to induce an exploration strategy using curiosity-driven methods which provide an additional pseudo reward for exploration in the environment [17]. However, such methods require carefully designed intrinsic rewards which can introduce expert-specific bias to the learning systems hindering the overall performance. Similarly, demonstration-guided methods have a high dependence on expert supervision which could be difficult to obtain in practice for many outdoor navigation tasks [18]. Instead of modifying the rewards, a more pragmatic approach is to

---

[*]These authors contributed equally to this work. Kasun Weerakoon, Souradip Chakraborty, Nare Karapetyan, Adarsh Jagan Sathyamoorthy, Amrit Singh Bedi and Dinesh Manocha are with the University of Maryland, College Park, MD, USA. Email: {kasunw, schakra3, knare, asathyam, amritbd, dmanocha} @umd.edu.

6th Conference on Robot Learning (CoRL 2022), Auckland, New Zealand.

try to work directly with the sparse rewards [19] and make the DRL based policies to train for the outdoor navigation task at hand. But the major challenge is how to induce inherent exploration into the training without reshaping the rewards. Recent work in literature[20, 21] suggest that one way to handle these is to use heavy-tailed policy (such as Cauchy) parametrization techniques. Motivated by these factors, in this work, we try to find optimal behaviors for outdoor navigation tasks while directly operating under sparse reward settings.

**Main Contributions:**

1. We present a heavy-tailed policy formulation with adaptive gradient tracking to address three navigation challenges encountered in complex outdoor environments (goal-reaching, obstacle avoidance, and navigating on uneven terrains). Our proposed algorithm HTRON outperforms three state-of-the-art algorithms: Reinforce [22], PPO [23], and TRPO [24] in terms of cumulative reward return with improved sample efficiency for faster convergence.

2. We propose a set of novel sparse reward functions that do not require careful reward designs yet are capable of successfully navigating a differential drive robot using an improved stochastic policy gradient method.

3. We evaluate the navigation performance of HTRON in both simulated and real-world complex outdoor environments using a Clearpath Husky robot. HTRON results in an increase of up to $34.41\%$ in terms of navigation success rate, $15.15\%$ decrease in the average time steps taken to reach the goal, and $24.9\%$ decrease in the elevation cost compared to the navigation policies obtained by the other methods.

## 2 Related Work

Learning-based approaches, in particular DRL have pushed the boundaries of mobile robot navigation for unstructured and dynamically changing environments [25, 26, 8]. Nevertheless, these methods suffer from sample efficiency under sparse rewards settings because dense rewards play an important role in learning for DRL methods. We summarize the related work next.

**Outdoor Robot Navigation:** Navigation, path planning and obstacle avoidance are extensively-studied fundamental research problems in the field of robotics [27, 28]. Number of deterministic [29, 30] and stochastic [31, 32] algorithms have been proposed in the past three decades. However, robot navigation in complex outdoor environments still remains a challenging problem given the complexity and diversity of the setting. To perform robot-centric decision-making, deep reinforcement learning (DRL) strategies have been employed in recent studies for outdoor navigation [33, 34]. For example, deterministic policies such as DDPG[26], A3C[35], and DQN[36] with dense rewards have been incorporated for robot navigation on uneven terrains. An adaptive model that directly learns control and environmental dynamics is presented in [8]. Moreover, segmentation[37] and self supervised [25] methods for identifying navigable regions have also been utilized with a combination of DRL based navigation methods to ensure stable and efficient navigation.

**Sparse Reward Settings:** In sparse reward settings, an environment rarely produces a useful reward which significantly affects the reinforcement policy leaning [11]. To deal with this issue, several reward shaping methods based on intrinsic curiosity and information gain-based shaping have been proposed [38]. For example, reward shaping based methods such as [39, 17] encourage the agent to explore unvisited states by modifying the actual reward output. However, such approaches require additional effort and expertise in reward function design. Imitation learning and learning from demonstration methods are another approach for dealing with sparse rewards in practice [40]. However, obtaining reliable expert demonstrations for continuous control tasks is practically infeasible and challenging [41, 15].

**Heavy-tailed Policy Parametrization:** The possibility of incorporating heavy-tailed distributions to RL tasks is theoretically analyzed in recent literature [42, 21]. For example, in [43], beta distribution is utilized with dense rewards for stochastic policy training to enhance the state exploration capabilities. Further, a heavy-tailed policy gradient method is proposed in [42] to find convergence to global maxima while minimizing the risk of local maxima convergence. Inspired by these ideas, we propose a heavy-tailed adaptive reinforce algorithm to deal with sparse rewards in complex outdoor environments.

# 3 Problem Formulation and Our Approach

## 3.1 Outdoor Navigation via Reinforcement Learning

Mathematically, we formulate the outdoor navigation problem as a Markov Decision Process (MDP) in continuous state and actions spaces:

$$\mathcal{M} := \{\mathcal{S}, \mathcal{A}, \mathbb{P}, r, \gamma\}, \tag{1}$$

where $\mathcal{S}$ denotes the state space including distance from the goal, heading, roll and pitch angle; $\mathcal{A}$ is the actions space (linear, angular velocity); $\mathbb{P}(s'|s,a)$ is the transition kernel; $r(s,a)$ is the reward and $\gamma \in (0,1)$ denotes the discount factor. The objective of the robot is to learn a navigation policy $\pi_\theta(a|s)$ parameterized by $\theta$ (which controls the probability of taking a particular action $a$ in given state $s$) to perform efficient outdoor navigation. This is achieved by solving the following optimization problem

$$\max_\theta J(\theta) := V^{\pi_\theta}(s_0), \tag{2}$$

where $V^{\pi_\theta}(s_0) = \mathbb{E}\big[\sum_{t=0}^\infty \gamma^t r(s_t, a_t) \mid s_0 = s, a_t \sim \pi_\theta(\cdot|s_t)\big]$ is the average cumulative reward (or value function), and $s_0$ denotes the initial state along a trajectory $\{s_t, a_t, r(s_t, a_t)\}_{u=0}^\infty$. We note that since the goal of the outdoor navigating robot is to reach a goal and to make sure that the problem (2) is actually achieving that, we need to design the reward function $r(s, a)$ such that upon maximizing its cumulative value, our goal is achieved. This is one of the major challenges in applying RL ideas to outdoor navigation. The majority of the RL results revolve around learning in standard environments and platforms like Gym, Mujoco [44] or other simulated environments where the rewards structures are already defined. However, in practical outdoor navigation scenarios, it is extremely hard to generate dense rewards due to the possibility of huge unknown trajectories a robot can take while navigating in the environment [34, 8]. The problem is exacerbated in continuous state-action spaces where one needs to define rewards even for infinite possibilities, which could be impractical.

## 3.2 Our Approach

To deal with this issue and empower the application of RL to outdoor navigation, we take a different route and advocate the use of sparse rewards. The advantage of sparse rewards is the simplicity of their design as they need to be defined only at specific goals/sub-goals to the robot. However, the sparse reward makes the learning problem hard due to the non-trivial estimation of value functions over the continuous state space, which can be mitigated through the use of heavy-tailed policy gradient algorithms[42, 21]. In this work, we start with a construction of an experimental sparse reward design methodology for outdoor navigation scenarios and proceed with the description of our algorithm.

### 3.2.1 Outdoor Navigation using Sparse Rewards

We categorize the level of complexity in the reward structure using three navigation scenarios in outdoor environments. In all scenarios, our agent is modeled as a differential drive robot with a two-dimensional continuous action space $a = (v, \omega)$ (i.e. linear and angular velocities), whereas the state space dimension varies for each scenario. The state inputs are obtained in real-time from the robot's odometry, and LiDAR sensors. The list of state inputs used throughout the three scenarios is as follows:

- $d_{goal} \in \mathbb{R}^+$ - current distance between the robot and its goal;
- $\alpha_{goal} \in [0, \pi]$ - current angle between the robot's heading direction and the goal;
- $(v_{t-1}, \omega_{t-1}) \in [-1, 1]$ - actions from the previous time step (i.e. linear and angular velocities);
- $(\theta_{roll}, \theta_{pitch}) \in [0, \pi]$ - roll and pitch angle of the robot respectively;
- $D_{obs} \in [0, 10]$ - laser scan vector that includes distance to the obstacles around the robot (i.e. $360°$ laser scan data as a vector with $\sim 720$ elements) at a given time.

**Scenario 1 (Goal Reaching Baseline):** In this scenario, the robot is placed in an obstacle-free outdoor terrain with an objective to reach a given goal location. We incorporate distance to the goal, heading angle to the goal and previous actions to define a four-dimensional state space $s =$

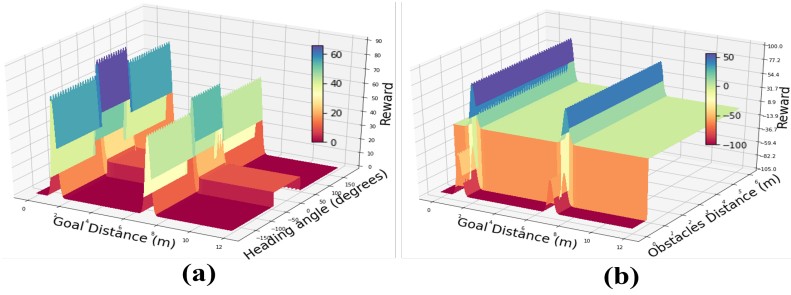

**(a)**                                           **(b)**

Figure 1: **Sparse reward surface visualization:** Reward distribution with (a) $d_{goal}$, $\alpha_{goal}$ and (b) $d_{goal}$, $D_{obs}$. We observe that the sparse reward settings lead to chaotic and unstructured reward surfaces. These plots show that our reward formulations in the navigation scenarios are significantly sparse and do not include uniquely identifiable maxima that can be reached smoothly.

$[d_{goal}, \alpha_{goal}, v_{t-1}, \omega_{t-1}]$. We utilize a set of sparse rewards that are easy to implement for policy training. For the goal reaching task, we define $r_{heading}$ and $r_{dist}$ rewards to maintain the robot's heading direction towards the goal and to encourage reaching the goal respectively. Hence,

$$r_{heading} = \mathbb{1}_{\{|\alpha_{goal}| \leq \pi/4\}}, \text{ and } r_{dist} = \frac{\beta_g}{2}\mathcal{N}(\frac{d_{goal}}{2}, \sigma_g^2) + \beta_g\mathcal{N}(0, \sigma_g^2), \tag{3}$$

where $\beta_g$ is a constant used to adjust the reward amplitude for goal reaching, $\sigma_g \in [0, 0.1]$ is the variance of the normal distribution $\mathcal{N}(., \sigma_g^2)$ which we maintain at a very low range. This ensures, the robot receives a reward $r_{dist}$ only when it reaches either $\frac{d_{goal}}{2}$ or the defined goal. Otherwise, $r_{dist}$ is zero. These reward formulations intuitively motivate the robot to reach the goal by rewarding it when the halfway or the complete distance to the goal is traveled. The total reward is obtained as, $r_{tot} = r_{heading} + r_{dist}$. The reward distribution w.r.t the state parametric space is visualized in Figure 1(a) to highlight the reward sparseness.

**Scenario 2 (Obstacle Avoidance):** In this scenario, the robot is placed in an outdoor area with static obstacles such as trees, walls, and sharp hills. The objective is to reach the goal while avoiding collisions with these obstacles. Additionally laser scan vector is taken into account in the state space $s = [d_{goal}, \alpha_{goal}, v_{t-1}, \omega_{t-1}, D_{obs}]$. In addition to the two rewards defined in the previous scenario (i.e. $r_{heading}$ and $r_{dist}$), we introduce an additional sparse reward $r_{obs}$ as,

$$r_{obs} = \begin{cases} -100 & \text{if } \min(D_{obs}) \leq d_{collision}, \\ 0 & \text{otherwise}, \end{cases} \tag{4}$$

to penalize collisions. Here, $d_{collision}$ is the minimum safety clearance that the robot should maintain with obstacles. The modified total reward is obtained as, $r_{tot} = r_{heading} + r_{dist} + r_{obs}$.

**Scenario 3 (Navigating on Uneven Terrains):** In this scenario, the robot is placed in a highly elevated outdoor terrain where it might flip over or experience instability in some regions. The objective is to navigate to a goal location while avoiding steep elevations in the environment. The corresponding state space is $s = [d_{goal}, \alpha_{goal}, v_{t-1}, \omega_{t-1}, \theta_{roll}, \theta_{pitch}]$. To minimize navigating on steep elevations, we incorporate another sparse reward term ($r_{stable}$) with the goal reaching reward pair. We consider the states where the robots roll ($\theta_{roll}$) or pitch angle ($\theta_{pitch}$) exceeds $\pm\frac{\pi}{4}$ as steep elevations (i.e. unstable robot orientations). Hence, the stability sparse reward is defined as,

$$r_{stable} = \mathbb{1}_{\{|\theta_{roll}| \geq \pi/4\}} \cup \mathbb{1}_{\{|\theta_{pitch}| \geq \pi/4\}}. \tag{5}$$

Hence, the total reward function for this scenario is obtained as, $r_{tot} = r_{heading} + r_{dist} + r_{stable}$.

### 3.2.2 Heavy-Tailed Reinforce Algorithm for Outdoor Navigation (`HTRON`)

Just operating with sparse rewards is not sufficient to solve the outdoor navigation problem because the learning procedure won't be able to explore the environment in a required manner. For instance, continuous control robotics research primarily relies on Gaussian policy parametrization given by $\pi_\theta(a|s) = \mathcal{N}(a|\varphi(s)^\top\theta, \sigma^2)$ where $\theta$ controls the mean of the Gaussian, $\varphi(s)$ denotes the states feature representation $\varphi : \mathcal{S} \rightarrow \mathbb{R}^d$ with $d \ll q$, and $\sigma^2$ is variance. However, the performance of Gaussian policy parametrization under sparse rewards is shown to suffer badly in the latest research

---

**Algorithm 1** Heavy-Tailed Reinforce for Outdoor Navigation (HTRON)

---
1: **Initialize** : Initial policy network parameter $\theta = \theta_0$, discount factor $\gamma$, step-size $\eta$, Adaptive momentum parameters $\beta_1$, $\beta_2$, $\epsilon$, $\delta$, and $\phi$
   **Repeat for** $k = 1, \ldots$
2: Collect the trajectory $\xi_k(\theta_k)$ by utilizing policy $\pi_{\theta_k}$ with robotic action driven constraints i.e the actions are first sampled as $a \sim \pi_{\theta_k}(a|s)$ and then we evaluate projections $\mathcal{P}_\infty(a)$ such that $\|a\|_\infty \leq \delta$
3: Estimate $\nabla J(\theta_k, \xi_k(\theta_k))$ (cf. (7)) and then perform clipping such that $\|\nabla J(\theta_k, \xi_k(\theta_k))\|_\infty \leq \phi$
4: Estimate $g_k$ and $\theta_{k+1}$ with adaptive moment estimation method using (8)
5: $k \leftarrow k + 1$
   **Until Convergence**
6: **Return:** $\theta_k$

---

[42, 21]. The primary reason for failure is due to the light-tailed nature of Gaussian distribution which restricts the policy to take action closer to its mean value and thereby restricts exploration. Typical methods to deal with exploration includes adding intrinsic curiosity [45] or entropy based exploration [17] which either requires learning the global dynamics model or estimation of the occupancy density function which is extremely expensive for continuous control robotics problem.

In this work, we leverage an alternate approach motivated by the latest research by [42, 21] and focus on developing a heavy-tailed parametrization based policy for outdoor navigation problems with additional robotics-driven constraints as detailed in Algorithm 1. We parameterize the policy by a heavy-tailed Cauchy distribution given by

$$\pi_\theta(a|s) = \frac{1}{\sigma\pi(1 + ((a - \varphi(s)^\top\theta)/\sigma)^2)}, \tag{6}$$

where $\sigma$ is the fixed variance. Now, finally we write the stochastic policy gradient [20] as

$$\nabla J(\theta_k, \xi_k(\theta_k)) = \sum_{t=0}^{T_k} \gamma^{t/2} r(s_t, a_t) \cdot \left( \sum_{\tau=0}^{t} \nabla \log \pi_{\theta_k}(a_\tau|s_\tau) \right), \tag{7}$$

where $\nabla J(\theta_k, \xi_k(\theta_k))$ denotes the unbiased estimator of gradient $\nabla J(\theta_k)$ at $\theta_k$, and $T_k \sim$ Geom$(1 - \gamma^{1/2})$. Although heavy-tailed parametrization induces exploration which has proven to be extremely beneficial in sparse reward settings [42, 21], it also induces instability in behaviour due to the high probability of taking extreme action even near optimal regions. Hence, [42] proposed gradient tracking to mitigate the above issue. However, instead of momentum tracking with mirror ascent type update as proposed in [42], we use an adaptive moment estimation based method as an optimizer with gradient clipping to stabilize the instability induced owing to the heavy-tailed parametrization which improves the time taken by gradient tracking based methods.

$$m_k = \beta_1 m_{k-1} + (1 - \beta_1)g_k, \tag{8}$$

$$v_k = \beta_2 v_{k-1} + (1 - \beta_2)g_k^2, \tag{9}$$

$$\theta_{k+1} = \theta_k + \frac{\eta}{\sqrt{v_k} + \epsilon}m_k, \tag{10}$$

where $m_k$, $v_k$ are the first and second moments respectively and $\beta_1$, $\beta_2$, $\epsilon$ are hyperparamters for the optimizer and $g_k = \nabla J(\theta_k, \xi_k(\theta_k))$ with the added constraint of clipping given as $\|\nabla J(\theta_k, \xi_k(\theta_k))\|_\infty \leq \phi$. Another important challenge is to incorporate practical constraints into the learning system which is specific to robotics based problems where the action is bounded. To solve the same, we use a simple projection $\mathcal{P}_\infty(\cdot)$ based technique where we project the output from the policy $\pi_\theta(a|s)$ into an infinite norm ball which constraints the action space within the bound and can be shown as $a \sim \pi_\theta(a|s)$ such that $\|a\|_\infty \leq \delta$.

## 4   Experiments

We conduct a detailed analysis and performance comparison of our proposed method against three state-of-the-art stochastic policy gradient algorithms: TRPO [24], PPO [23], and Reinforce [22]. We used identical policy distribution parameters (e.g. distribution standard deviation $\sigma$) and neural network architectures with all four algorithms during training and evaluation.The heavy-tailed policies

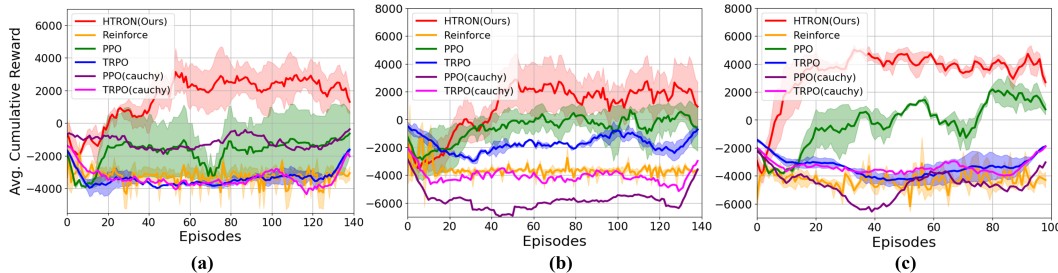

Figure 2: Performance comparison of `HTRON` with Reinforce [22], PPO [23], TRPO [24], PPO (with Cauchy) and TRPO (with Cauchy) in three scenarios: (a) Goal-reaching Baseline, (b) Obstacle avoidance, and (c) Navigating on uneven terrains. Each episode contains 300 times steps (i.e. $\sim 30k$-$42k$ overall time steps). All the policies are trained with a fixed standard deviation $\sigma = 0.25$. We note that `HTRON` achieves the highest average reward returns. The plots are averaged over 6 random seeds and shows mean and confidence intervals.

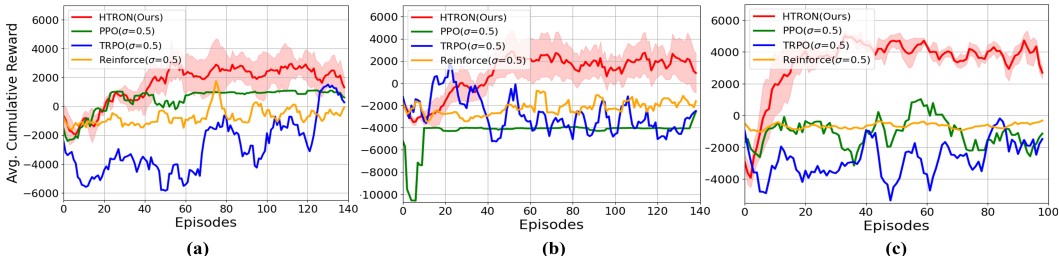

Figure 3: Performance comparison of `HTRON` with Reinforce [22], PPO [23], and TRPO [24] policies with large standard deviations ($\sigma = 0.5$) in three scenarios: (a) Goal-reaching Baseline, (b) Obstacle avoidance, and (c) Navigating on uneven terrains. We observe that the Gaussian policies cannot achieve faster policy convergence by simply increasing the standard deviation of the policy distribution. Instead, the Gaussian policies demonstrate higher instability during training.

are implemented and trained with Pytorch using a Unity based realistic outdoor simulator, Clearpath Husky robot model, and ROS Melodic platform. The Unity simulator includes diverse terrains and elevations for training and testing. Further, we deploy `HTRON` on a real Clearpath Husky robot on outdoor terrains.

## 4.1 Policy Learning Performance

We evaluate the performance of all algorithms through the policy convergence rate (see Figure 2). In the goal-reaching baseline scenario, we observed that standard Gaussian policy-based algorithms cannot achieve the highest reward return when we incorporate the sparse rewards. In contrast, our heavy-tailed policy is able to obtain maximum rewards in a sample-efficient manner as shown in Figure 2(a). This is primarily due to the improved exploratory behavior achieved by the heavy-tailed policy in comparison to the standard Gaussian policies.

In Scenario 2 with static obstacles, our experiments showed that the random and sparse obstacle penalties significantly affect policy convergence and stability. In particular, PPO converges to the halfway point towards the goal with multiple collisions. However, our method successfully learns policies to reach the goal within $\sim 40$ episodes (see Figure 2(b)). This demonstrates the accelerated learning capabilities provided by the heavy-tailed formulation compared to the traditional stochastic policy gradient methods. In Scenario 3, we include $r_{stable}$ to penalize the actions that could navigate the robot towards steep elevations. We observe that Reinforce and TRPO based policies struggle to even reach the goal under this setting while PPO manages to reach the goal in only a few episodes without proper convergence.

Further, we performed additional experiments on PPO, TRPO, and Reinforce algorithms by increasing the standard deviation ($\sigma = 0.5$) of their Gaussian policy distribution(see Fig. 3). We observe that even with increasing variance, our algorithm (HTRON) outperforms all the baselines including PPO, TRPO, Reinforce, etc which demonstrates the superiority of our algorithm. Additionally, We observe that the Gaussian policies cannot achieve faster or better policy convergence by simply in-

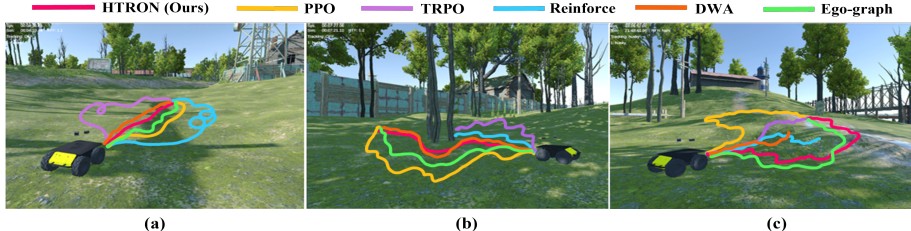

**Figure 4:** Trajectories when navigating on different simulated outdoor terrains using HTRON, PPO[23], TRPO[24], Reinforce[22], DWA[30] and Ego-graph[46, 47]: (a) Goal-reaching baseline, (b) Obstacle avoidance environment with multiple trees, walls, and unreachable steep elevations. (c) Uneven terrain environment with different levels of elevations and flat terrains. HTRON takes the shortest time steps to successfully reach the goal while maintaining the lowest elevation cost on uneven terrains.

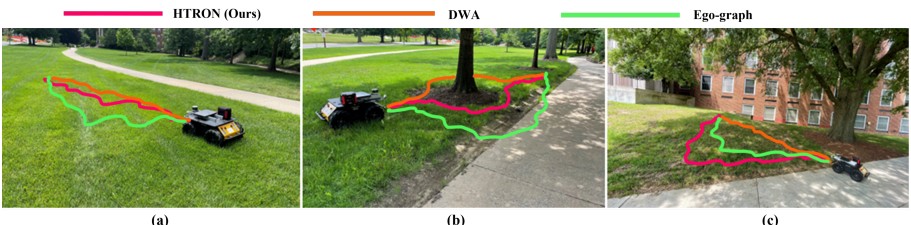

**Figure 5: Navigating in real world using HTRON:** (a) Goal-reaching scenario, (b) Obstacle avoidance. (c) Uneven terrain navigation. We deploy our algorithm on Clearpath Husky robot demonstrating the ease of transferring HTRON on real systems without significant performance degradation. The robot is equipped with a VLP16 LiDAR, and a laptop with an Intel i9 CPU and an Nvidia RTX 2080 GPU.

creasing the standard deviation of the policy distribution. Instead, the Gaussian policies demonstrate higher instability during training. Hence, we restricted to $\sigma = 0.25$ as presented in Fig. 2 during our experiments.

## 4.2 Navigation Performance

We compare our method's navigation performance qualitatively in Figure 4 and quantitatively in Table 1. The quantitative evaluations of the navigation trajectories are based on the following metrics:

- **Success Rate:** The percentage of successful goal reaching attempts without any collisions or flip-overs out of the total experiments.
- **Avg. Trajectory Length:** The average number of time steps taken by the robot to reach the goal.
- **Elevation Cost:** Norm of the elevation gradient experienced by the robot throughout a trajectory (i.e. $||\nabla z_r||$, where $z_r$ is the vector that includes vertical motions of the robot's along a trajectory).

We would like to note that in this work rather than proposing a competitive navigation algorithm, the primary objective is to highlight the importance of heavy-tailed distributions for faster policy convergence under practical sparse reward settings. However, in addition to comparing against the DRL-based methods, we evaluate the navigation performance of two classical algorithms: Dynamic Window Approach(DWA)[30] and Ego-graph[47]) under the same test conditions. We observe that DWA and Ego-graph outperform all the DRL based methods in terms of success rate during goal reaching and obstacle avoidance. However, HTRON demonstrates comparable or better performance during uneven terrain navigation in terms of all the evaluation metrics.

We further notice that our method maintains the highest success rate in all three scenarios while other DRL policies demonstrate a significantly low performance in successful goal-reaching. This is primarily due to the poor convergence in Gaussian policies under sparse reward settings. Further, HTRON reaches the goal faster using fewer time steps than the other three methods. This indicates that our policy has optimized to collect more goal completion rewards by reaching the goal location quickly. Policy convergence plots presented in Figure 2 further validates this argument.

Navigation scenarios 1 and 2 are common in both indoor and outdoor environments. Hence, we further investigate the importance of our algorithm to handle uneven terrains encountered in complex outdoor environments. We observe that our method is capable of avoiding steep elevations from the

| Scenario | Algorithm | Success Rate (%)↑ | Avg. Trajectory Length (# time steps)↓ | Elevation Cost (m)↓ |
|---|---|---|---|---|
| **Goal Reaching Baseline** | TRPO[24] | 32.28 | 236 | 0.764 |
| | PPO[23] | 45.76 | 198 | 0.648 |
| | Reinforce[22] | 36.91 | 207 | 0.655 |
| | DWA[30] | **100** | 172 | 0.645 |
| | Ego-graph[47] | 88.86 | 186 | **0.637** |
| | HTRON(Ours) | 72.46 | **168** | 0.641 |
| **Obstacle Avoidance** | TRPO[24] | 18.74 | 243 | 0.637 |
| | PPO[23] | 37.56 | 224 | 0.652 |
| | Reinforce[22] | 8.93 | 286 | **0.624** |
| | DWA[30] | **100** | 218 | 0.645 |
| | Ego-graph[47] | 58.79 | 238 | 0.642 |
| | HTRON(Ours) | 62.21 | **209** | 0.639 |
| **Navigating on Uneven Terrains** | TRPO[24] | 25.87 | 273 | 1.842 |
| | PPO[23] | 37.46 | 264 | 1.674 |
| | Reinforce[22] | 18.33 | 270 | 1.336 |
| | DWA[30] | 27.38 | 268 | 1.943 |
| | Ego-graph[47] | 65.89 | 198 | 1.316 |
| | HTRON(Ours) | **71.87** | **244** | **1.257** |

Table 1: **Navigation Comparisons:** HTRON outperforms the other DRL methods in terms of success rate and the average time to reach the goal. It also maintains the lowest elevation cost when navigating on uneven terrains. Reinforce and TRPO obtain slightly lower elevation costs during goal-reaching and obstacle avoidance scenarios by rotating around flat regions without reaching the actual goal.

changing roll and pitch angles before leading to robot flip-overs (see Figure 2(c) and 4(c) ). Hence, trajectories generated by our algorithm significantly minimize the elevation cost experienced by the robot while other methods navigate along steep terrains with higher elevation costs.

Finally, we integrate our algorithm into a real Clearpath Husky robot to demonstrate navigation capabilities in real outdoor settings. We observe that our method can successfully perform the navigation tasks we trained in the simulator. Sample navigation trajectories from three real outdoor scenarios are presented in Figure 5.

## 5 Limitations and Analysis

We discuss the limitations of our method that we expect to address in the future. The robot cannot identify the uneven terrains without experiencing the uneven region due to the orientation based elevation sensing. A 3D LiDAR pointcloud based elevation map could be utilized in state space to identify the elevation changes in the robot's vicinity without explicitly experiencing elevation changes from the orientation data. Moreover, the heavy-tailedness of the policy distribution could lead to convergence instabilities. Especially, we observe considerable instability and non-converging behavior when we try to use PPO and TRPO with heavy-tailed distributions. Hence, further investigations need to be conducted to develop robust and stable heavy-tailed policy models for higher dimensional state spaces and more complex sparse reward settings.

## 6 Conclusion

In this work, we presented a novel approach for adapting heavy-tailed policy based control strategy for efficient outdoor mobile robot navigation – HTRON. The proposed method is able to overcome the major issue of sparse rewards inherent to DRL based approaches. We show that without hand-crafted reward shaping methods we are able to accelerate learning capabilities and produce a reliable navigation policy. We have assessed the performance of navigation policies in three different scenarios based on their complexity and corresponding objectives. All policies have been trained in a ROS-based high-fidelity Unity simulator. We report a performance comparison of our method against REINFORCE, PPO, and TRPO algorithms in terms of learning rate and navigation quality. We also deployed HTRON on real Clearpath Husky in outdoor terrains. And finally, we report the limitations, possible improvements, and future work that this approach can lead to robust outdoor navigation.

**Acknowledgments**

This work was supported in part by ARO Grants W911NF1910069, W911NF2110026, and Army Cooperative Agreement W911NF2120076. We acknowledge the support of the Maryland Robotics Center.

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
