# OpenReview forum: "HTRON: Efficient Outdoor Navigation with Sparse Rewards via  Heavy Tailed Adaptive Reinforce Algorithm"
_robot-learning.org/CoRL/2022/Conference — CoRL 2022 Poster_

### Official Review · Reviewer_B9Ro · 2022-07-07

**Originality:** Good
**Technical Quality:** Good
**Clarity Of Presentation:** Very Good
**Impact:** 3

**Recommendation:**

Weak Reject: I recommend rejecting the paper, but will not argue for my recommendation if the majority of other reviewers have a different opinion.

**Summary:**

This paper investigates learning three navigation tasks with a husky robot, in simulation and with demonstrations on a real platform. Different from typical methods, this work considers the challenges standard RL algorithms have with sparse rewards, proposing a heavy-tailed approach to improve exploration. To improve learning stability, gradient tracking and clipping was required. The results indicate an increase in success rate for navigating to a goal compared with several baseline algorithms.

**Issues:**


Spelling / Grammar / Typos:
“Number of deterministic” - missing “A”

“For instance, majority of the research”

“restricts the policy to take action very”

“for outdoor navigation problem”

“which constraints the action space”

“evaluation.The”

“in outdoor terrains” - on outdoor terrains

“via utilizing policy”

“assessed the performance navigation policies”

“∥∇J(θ k , ξ k (θ k ))∞ ≤ ϕ” - Missing closing double pipes before infinity

“motions of the robot’s along a trajectory”

“deployed HTRON on real Clearpath Husky”

“However, instead of momentum tracking with mirror ascent type update as proposed in [42], we use an adaptive moment estimation based method as an optimizer with gradient clipping to stabilize the instability induced owing to the heavy-tailed parametrization which improves the time taken by gradient tracking based methods.” - Difficult sentence, missing hyphens and probably a comma or two.


**Quality Of The Limitations Section:**

Additional details required

**Reviewer Expertise:**

4: The reviewer is confident but not absolutely certain that the evaluation is correct

**Robotics Focus:**

Sufficient demonstration on hardware

**Strengths And Weaknesses:**

This paper was well written, presented clearly, and easy to follow.

The primary concern is that the contribution is minor. The paper focuses heavily on the three elected navigation tasks, however, these seem relatively simple problems for reinforcement learning to solve if the imposed sparse reward limitation was lifted, particularly as the ground-truth goal information (non-sparse, distance and angle to the goal) is provided as input to the policy.

 It seems a constant variance is used throughout for sampling stochastic actions. Does the choice of variance affect learning, and what effect would having variance as a learnable parameter have on performance? This is particularly interesting for the comparison between Cauchy and Gaussian distributions for exploration. The resulting policies produce non-smooth trajectories, suggesting that more improvements are needed.


**Summary Of Recommendation:**

One way to make this method more appealing would be to show quantitative learning improvements with a sparse reward over a shaped reward, i.e. show that this task benefits from a sparse reward, and that this method improves learning with such a reward. For now, the reader needs to imagine that a continuous/shaped reward is difficult to design, which is not the case for the presented problem. For this, a different task setup may need to be investigated.

Given the outcome of this paper is improved exploration for learning with sparse rewards, it is surprising that the Soft Actor-Critic algorithm is missing, from the comparisons and from the extensive reference list. Inclusion, or a discussion on reasons for exclusion, would be useful.

An ablation of some of the techniques and design choices, for example, the effect of gradient tracking and clipping, and choice of variance, would be useful for the community to understand what is difficult about using heavy-tailed policies.

---

> ### Author Response · Authors · 2022-08-21
> **Response to Reviewer B9Ro**
>
> We thank the reviewer for appreciating the clarity of our paper and recommending corrections to the Spelling/Typos. The detailed replies are as follows.
>
> >**Weakness 1:** It seems a constant variance is used throughout for sampling stochastic actions. Does the choice of variance affect learning, and what effect would having variance as a learnable parameter have on performance? This is particularly interesting for the comparison between Cauchy and Gaussian distributions for exploration. The resulting policies produce non-smooth trajectories, suggesting that more improvements are needed.
>
> **Reply to Weakness 1:** We have conducted experiments on PPO, TRPO, and Reinforce learning performance with larger variances. Please see the comparisons in Fig. 3 in the revised version of the paper. We observe that increasing the variance cannot significantly improve the learning capabilities of Gaussian policies. For instance, increasing the variance of Gaussian would actually mean, that we are increasing the probability of taking actions away from the mean, but it would still not make sure that the probabilities of taking extreme actions are high enough. Also, making exploration depending upon variance would also result in an additional hyperparameter in the problem which we would like to avoid in practice.
>
>
> >**Weakness 2:** One way to make this method more appealing would be to show quantitative learning improvements with a sparse reward over a shaped reward, i.e. show that this task benefits from a sparse reward, and that this method improves learning with such a reward. For now, the reader needs to imagine that a continuous/shaped reward is difficult to design, which is not the case for the presented problem. For this, a different task setup may need to be investigated.
> Given the outcome of this paper is improved exploration for learning with sparse rewards, it is surprising that the Soft Actor-Critic algorithm is missing, from the comparisons and from the extensive reference list. Inclusion, or a discussion on reasons for exclusion, would be useful.
>
> **Reply to Weakness 2:** The actor critic algorithm is just an advanced version of the reinforcement algorithm where we utilize a critic to estimate the q value we use in reinforce, we restrict our comparisons to policy gradient based algorithms only. This is mainly because our contribution is related to exploring a fundamental idea of utilizing heavy tailed policy for continuous control tasks, this paper is actually the evidence that this works in practice, rather than actually claiming that the proposed algorithm is the best among all available methods in the literature. That we leave to the future scope of this work.
>
>
>
> >**Weakness 2:** An ablation of some of the techniques and design choices, for example, the effect of gradient tracking and clipping, and choice of variance, would be useful for the community to understand what is difficult about using heavy-tailed policies
>
> **Reply to Weakness 2:**
> Our contribution to this work is to show that using heavy tailed policy in DRL algorithms instead of Gaussian policies is outperforming in terms of exploration quality and the learning rate. We have included several additional experiment results to showcase that the Gaussian policies cannot achieve better performance by simply tuning distribution parameters such as variance (see Fig. 3 in the updated version of the paper uploaded with the comment).

---

### Official Review · Reviewer_skdZ · 2022-07-26

**Originality:** Fair
**Technical Quality:** Fair
**Clarity Of Presentation:** Good
**Impact:** 2

**Recommendation:**

Weak Accept: I recommend accepting the paper, but will not argue for my recommendation if the majority of other reviewers have a different opinion.

**Summary:**

Given prior work establishing that heavy-tailed policies help with exploration for sparse reward RL both in theory [20] and practice [21], this work explores evaluating heavy-tailed policies for sparse reward outdoor navigation. Since heavy tailed policies have already been experimentally studied for sparse reward RL in [21], the primary contribution of this paper appears to be the application of this same idea to outdoor navigation problems. The main differences of HTRON from the approach presented in [21] appear to be (1) the use of an adaptive moment estimation method to stabilize policy updates instead of the moment tracking approach used in [21] and (2) an extra projection step to handle bounded action spaces.

**Issues:**

The core issue with this paper is it not clear to me what the contribution is relative to [21]. If the contribution is a heavy-tailed RL algorithm specifically designed for navigation problems, then I would expect a lot more structure specific to navigation to be built into HTRON. If the contribution is instead a generally more effective heavy-tailed RL algorithm, then the authors must argue what the benefits of HTRON are relative to [21] and illustrate these benefits through experimental comparison.

**Quality Of The Limitations Section:**

Limitations are addressed clearly

**Reviewer Expertise:**

4: The reviewer is confident but not absolutely certain that the evaluation is correct

**Robotics Focus:**

Sufficient demonstration on hardware

**Strengths And Weaknesses:**

Strengths:

(1) The setting of outdoor navigation is definitely an important one in robotics as a whole, and is a setting where sparse reward RL is very well motivated.

(2) HTRON is thoroughly evaluated on a suite of simulation tasks and a physical outdoor navigation task.

Weaknesses:

(1) The methodological contribution of this paper is not clear to me. The only differences of HTRON from the approach presented in [21] appear to be (1) the use of an adaptive moment estimation method to stabilize policy updates instead of the moment tracking approach used in [21] and (2) an extra projection step to handle bounded action spaces. If the claim is that HTRON is an heavy-tailed RL algorithm specifically designed for navigation problems, then I would expect more structure specific to outdoor navigation to be built into HTRON. Without this, it is hard to understand what HTRON contributes beyond [21] since the impact of the adaptive moment estimation method is not evaluated in experiments and the handling of bounded action spaces is a very minor addition. Indeed, it is strange that no direct comparison between [21] and HTRON is performed given how heavily HTRON is based on [21].

(2) The experimental comparisons are also a bit unclear. From my understanding, the only difference between Reinforce and HTRON in Figure 2 is the use of a heavy-tailed policy in HTRON. If this is indeed correct, then the comparison to PPO and TRPO, which presumably use Gaussian policies is a bit strange. It would make more sense to compare PPO and TRPO to their equivalents if a heavy-tailed policy were used.

**Summary Of Recommendation:**

The methodological contribution of this paper seems unclear/incremental with respect to prior work on heavy-tailed policies for RL. While applying these prior works to navigation problems is interesting, this alone does not seem sufficient for publication at CoRL unless further evidence is provided (both written and empirical) to differentiate the proposed approach from prior algorithms.

Post Rebuttal:

I appreciate the authors' point though that one core contribution of the work is in the "design and analysis of the Sparse reward scenarios which are practical and relevant for Outdoor Navigation problems", which I missed in my initial review. The added comparison to classical algorithms though significantly improves my view of the thoroughness of the empirical evaluation, so I will raise my score to a Weak Accept once I am able to adjust my score. However, I still believe that comparison to [21] should be performed if possible.

---

> ### Author Response · Authors · 2022-08-21
> **Response to Reviewer skdZ**
>
> We thank the reviewer for appreciating the relevance and motivation of our paper. The detailed replies are as follows.
>
> >**Weakness 1:** The methodological contribution of this paper is not clear to me. The only differences of HTRON from the approach presented in [21] appear to be (1) the use of an adaptive moment estimation method to stabilize policy updates instead of the moment tracking approach used in [21] and (2) an extra projection step to handle bounded action spaces. If the claim is that HTRON is an heavy-tailed RL algorithm specifically designed for navigation problems, then I would expect more structure specific to outdoor navigation to be built into HTRON. Without this, it is hard to understand what HTRON contributes beyond [21] since the impact of the adaptive moment estimation method is not evaluated in experiments and the handling of bounded action spaces is a very minor addition. Indeed, it is strange that no direct comparison between [21] and HTRON is performed given how heavily HTRON is based on [21].
>
> **Reply to Weakness 1:** We thank the reviewer for pointing this out. We note that even though the idea of utilizing heavy-tailed policy parameterization is available in [21], the way we proposed to utilize it is different from the one presented in [21].
>
> Specifically, we remark that just changing the Gaussian policy parametrization by the Heavy tailed policy parameter does not help in practice and results in an unstable behavior of the proposed algorithm. In [21], a momentum-based approach (step 3 in Algorithm 1 in [21]) is used to get rid of this issue but it is difficult to implement that in practice because it requires stochastic policy gradients at two different iterates.
>
> Further, [21] requires the estimation of Bregman divergence which needs the inversion of the Hessian matrix which can be computationally expensive. On the other hand, we care more about the practical applicability of the approach and therefore propose to utilize adaptive moment estimation and gradient clipping in this work. So, this paper and [21] have fundamental differences.
>
> The practical applicability of the algorithm in [21] is not even discussed and also seems difficult because of the requirement of stochastic policy gradient at two different points, and hence not presented/compared in this work.
>
> Hence, we remark that an important aspect of the work lies in the design and analysis of the Sparse reward scenarios which are practical and relevant for Outdoor Navigation problems. We relate the Sparse reward function formulation with our HTRON algorithm and justify the performance improvement carefully, which is missing from the existing literature. We believe that this work is the first to design and analyze practical sparse reward functions for Outdoor Navigation Scenarios and propose a potential solution by relating to the reward surface.
>
>
>
> >**Weakness 2:** The core issue with this paper is it not clear to me what the contribution is relative to [21]. If the contribution is a heavy-tailed RL algorithm specifically designed for navigation problems, then I would expect a lot more structure specific to navigation to be built into HTRON. If the contribution is instead a generally more effective heavy-tailed RL algorithm, then the authors must argue what the benefits of HTRON are relative to [21] and illustrate these benefits through experimental comparison.
>
> **Reply to Weakness 2:** We have further included experimental and performance comparison results in terms of Policy training (see Fig. 2 and 3) and robot Navigation (see Fig. 4, 5 and Table 1).

---

### Official Review · Reviewer_N3Ka · 2022-08-01

**Originality:** Good
**Technical Quality:** Good
**Clarity Of Presentation:** Good
**Impact:** 3

**Recommendation:**

Weak Accept: I recommend accepting the paper, but will not argue for my recommendation if the majority of other reviewers have a different opinion.

**Summary:**

The authors propose HTRON, a deep reinforcement learning (DRL) framework intended to support outdoor robot navigation. Three problems of interest (goal-reaching, obstacle avoidance, navigating on uneven terrain) are configured to provide sparse rewards, and heavy-tailed policy parameterizations are utilized to induce exploration. HTRON repeatedly selects a control action for an unmanned ground vehicle (UGV) based on inputs that include distance and relative heading to goal, the robot’s attitude and previous control action, and the contents of a 360-degree laser scan. Compared with other DRL frameworks, HTRON is shown to converge more successfully to an effective navigation policy, and achieves superior success rates and navigation performance over the three representative tasks selected. Although trained in simulation, HTRON is also deployed successfully on a real UGV.

**Issues:**

- Please clarify how the outdoor navigation scenarios encountered in the training process differed (or did not differ) from the simulated scenarios encountered at test-time.

- Please also clarify the motivations for using DRL to solve the outdoor navigation problems explored in this paper, why only a single scan-line of the lidar is used (which also appears to be down-sampled from the available resolution of the sensor), and why no classical algorithms are included in the experimental comparison. The full 3D field of view of the lidar used in your experiments is likely capable of real-time terrain traversability mapping and path planning that could solve the problems of interest with comparable or superior performance to DRL.

**Quality Of The Limitations Section:**

Limitations are addressed clearly

**Reviewer Expertise:**

4: The reviewer is confident but not absolutely certain that the evaluation is correct

**Robotics Focus:**

Sufficient demonstration on hardware

**Strengths And Weaknesses:**

+ The paper addresses a topic relevant to robot learning, proposing a novel method for performing outdoor UGV navigation using deep reinforcement learning.

+ The paper is well-organized and clearly written, and the supplemental video provides very helpful illustrative examples of HTRON’s performance in simulation (alongside competing DRL algorithms) and aboard a real UGV.

+ HTRON is shown to address the three selected UGV navigation problems with superior rates of success relative to the selected DRL baseline algorithms, and offers superior navigation performance in nearly all instances as well.


- The prospect for high-performance outdoor UGV navigation is severely limited by the decision to work with a single scan line, rather than to fully utilize the 3D volumetric scanning lidar that the authors’ UGV appears to be equipped with. This sensor can be utilized without DRL to successfully achieve each of the three navigation tasks examined in this paper - none of these tasks represent unsolved problems that require the use of DRL.

- The paper would be strengthened by providing clarification on the extent to which the outdoor navigation scenarios encountered in the training process differed (or did not differ) from the simulated scenarios encountered at test-time.

- The paper would also be strengthened by discussing the various ways that classical algorithms (which do not invoke machine learning, and do not require a prior map of the terrain) are capable of solving the outdoor navigation problems discussed in this paper, and possibly also including them in the experimental comparison to motivate the use of DRL to solve these problems.

**Summary Of Recommendation:**

In HTRON, the authors have contributed a novel deep reinforcement learning framework for solving three outdoor navigation problems for lidar-equipped UGVs: goal-reaching, obstacle avoidance, navigating on uneven terrain. Impressively, the navigation policies learned in simulation outperform other DRL algorithms, and also transfer successfully to real robot hardware. However, the paper would be strengthened by providing stronger motivation and evidence for why DRL is needed to solve these problems.

Post-Rebuttal Comments: The authors' revisions and clarification are much appreciated, and the revisions have strengthened the paper. Taking these into account (especially the new results that have been added to the study) along with the CoRL review criteria, I have decided to elevate my rating to weak accept.

---

> ### Author Response · Authors · 2022-08-21
> **Response to Reviewer N3Ka**
>
>
> We thank the reviewer for appreciating the clarity, and relevance of our paper and for recommending valuable improvements. The detailed replies are as follows.
>
> >**Weakness 1:**  The prospect for high-performance outdoor UGV navigation is severely limited by the decision to work with a single scan line, rather than to fully utilize the 3D volumetric scanning lidar that the authors’ UGV appears to be equipped with. This sensor can be utilized without DRL to successfully achieve each of the three navigation tasks examined in this paper - none of these tasks represent unsolved problems that require the use of DRL.
>
> **Reply to Weakness 1:** We agree that several classical methods can be used to perform goal reaching and obstacle avoidance successfully using 3D Lidar data. However, navigating on uneven terrains is non-trivial to solve using classical methods even with a 3D Lidar. Especially the available 3D pointcloud based elevation mapping algorithms are relatively slow and inaccurate in cluttered environments [A, B]. Pointcloud based SLAM approaches such as LEGO-LOAM [C] are fast enough to generate 3D pointcloud maps. However, the ground elevation estimation accuracy requires higher-end lidars with more laser channels (32 channels or above).
>
> Nevertheless, we would like to highlight that in this work rather than proposing a competitive navigation algorithm, the primary objective is to highlight the importance of heavy-tailed distribution for faster policy convergence under practical sparse reward settings, which is not well explored in the existing literature.
>
> [A] P. Fankhauser, M. Bloesch, C. Gehring, M. Hutter, and R. Siegwart, "Robot-Centric Elevation Mapping with Uncertainty Estimates", in International Conference on Climbing and Walking Robots (CLAWAR), 2014.
>
> [B] P. Fankhauser, M. Bloesch, and M. Hutter, "Probabilistic Terrain Mapping for Mobile Robots with Uncertain Localization", in IEEE Robotics and Automation Letters (RA-L), vol. 3, no. 4, pp. 3019–3026, 2018.
>
> [C] T. Shan and B. Englot, "LeGO-LOAM: Lightweight and Ground-Optimized Lidar Odometry and Mapping on Variable Terrain," 2018 IEEE/RSJ International Conference on Intelligent Robots and Systems (IROS), 2018, pp. 4758-4765.
>
> >**Weakness 2:** The paper would be strengthened by providing clarification on the extent to which the outdoor navigation scenarios encountered in the training process differed (or did not differ) from the simulated scenarios encountered at test-time.
>
>
> **Reply to Weakness 2:**  We have utilized a unity based outdoor simulator that includes diverse terrains and elevations for training and testing. We restrict ourselves to a 100mx100m region during training. However, the real and simulated testing were conducted in different outdoor scenarios. Particularly, the real-world environments are reasonably different in terms of terrain structure(i.e. elevation gradient) and surface properties(i.e grass and tiny gravel regions with different levels of friction) which are not included or modeled in the simulator (e.g. elevation gain in the simulator is up to ~4m, however, we restricted it to ~3m elevation gain in the real world for robot’s safety)
>
> >**Weakness 3:** The paper would also be strengthened by discussing the various ways that classical algorithms (which do not invoke machine learning, and do not require a prior map of the terrain) are capable of solving the outdoor navigation problems discussed in this paper, and possibly also including them in the experimental comparison to motivate the use of DRL to solve these problems.
>
> **Reply to Weakness 3:** We have now included comparison results for two classical algorithms namely: Dynamic Window Approach (DWA) and Ego-graph. DWA only utilizes 2D scan data for obstacle avoidance (see Fig. 4, 5 and Table 1). Hence, it performs well during the goal reaching and obstacle avoidance tasks. However, navigating on elevated terrains is still a challenging problem in terms of both perception and planning for most of the classical algorithms.
> To this end, Ego-graph baseline method in [A] utilizes robot-centric elevation data to find the actions that minimize the elevation gradient cost. Hence the ego-graph planner is capable of navigating uneven terrains successfully. However, such methods' behavior is limited by the size of the action space which could lead to inconsistent navigation performance on complex uneven terrains.
> Further, the model based methods are generally restricted to a set of actions learned by the model. In contrast, end-to-end DRL methods are known for generating novel actions due to their inherent exploration capabilities [B].
>
> [A] S. Josef and A. Degani. Deep reinforcement learning for safe local planning of a ground401 vehicle in unknown rough terrain. IEEE Robotics and Automation Letters, 2020.
>
> [B]Dhiman, V., Banerjee, S., Griffin, B., Siskind, J.M. and Corso, J.J., 2018. A critical investigation of deep reinforcement learning for navigation. arXiv.

---

### Official Review · Reviewer_cxNk · 2022-08-03

**Originality:** Fair
**Technical Quality:** Fair
**Clarity Of Presentation:** Fair
**Impact:** 3

**Recommendation:**

Strong Accept: I recommend accepting the paper and will argue for my recommendation even if other reviewers hold a different opinion.

**Summary:**

This paper proposes a better policy parameterization that works better with sparse rewards in outdoor navigation tasks. The challenge of training RL agents in navigation tasks is reward sparsity since the rewards are only emitted upon the agent reaches the destination. Prior works use reward shaping/engineering to make training easier, but it requires intensive human efforts. The sparse reward function is easy to design but challenging to RL algorithms. The author believes sparse rewards make exploration difficult, and Gaussian policy used in policy gradient algorithms tends to over-commit to the mean actions. Thus, the author proposes to replace the Gaussian policy with a heavy-tailed distribution that has a higher chance of sampling actions away from the mean. The experimental results in both simulator and real-world show good performance.

**Issues:**


1. Line 182: what is gradient tracking? it would be better to briefly summarize how [42] do this.
2. "training instability" is used a lot in this paper but never formally defined.

**Quality Of The Limitations Section:**

Limitations are addressed clearly

**Reviewer Expertise:**

3: The reviewer is fairly confident that the evaluation is correct

**Robotics Focus:**

Sufficient demonstration on hardware

**Strengths And Weaknesses:**


## Strength
- The idea of using heavily tailed distribution to solve sparse rewards is sensible and elegant.

## Weakness
- Lack of experiments: If the argument is that sparse reward is preferred and HITRON can learn a good policy with sparse rewards, it would be necessary to compare PPO/TRPO/Reinforce with intensively tuned rewards. Otherwise, the reason of preferring HITRON over other baselines+tuned reawrds can be weak.
- Lack of comparison: If the primary benefit of HITRON is better exploration, I'm curious whether PPO/TRPO/Reinforce can match HITRON by increasing the standard deviation.

**Summary Of Recommendation:**

I recommend weak acceptance since this paper has a reasonable motivation and method, and the experiment supports the claim, despite lacking comprehensiveness. Overall, there are no obvious technical flaws. This makes this paper pass the bar of acceptance.

---

> ### Author Response · Authors · 2022-08-21
> **Response to Reviewer cxNk**
>
>
> **General Response:** We thank the reviewer for appreciating the contributions and recommending acceptance. The detailed replies are as follows.
>
> > Weakness: Lack of experiments: If the argument is that sparse reward is preferred and HITRON can learn a good policy with sparse rewards, it would be necessary to compare PPO/TRPO/Reinforce with intensively tuned rewards. Otherwise, the reason of preferring HITRON over other baselines+tuned reawrds can be weak.
> > Lack of comparison: If the primary benefit of HITRON is better exploration, I'm curious whether PPO/TRPO/Reinforce can match HITRON by increasing the standard deviation.
>
> **Reply to Weakness:** We have conducted experiments on PPO, TRPO, and Reinforce (with tuned rewards) algorithms by increasing the standard deviation of their Gaussian policy distributions. We observe that the increase in standard deviation cannot significantly improve the exploration capabilities of the Gaussian policy-based algorithms. Instead, the Gaussian policies sometimes demonstrate higher instability during training.
> We think better exploration is not always tied to high variance. For instance, increasing the variance of Gaussian would actually mean, that we are increasing the probability of taking actions away from the mean, but it would still not make sure that the probabilities of taking extreme actions are high enough. Also, making exploration depending upon variance would also result in an additional hyperparameter in the problem which we would like to avoid in practice.
> Also, an important point is to understand that PPO and TRPO both fall under trust-region-based policy gradient algorithms which are done using constrained policy optimization. In other words, the policies are constrained to be in a trust region and explore less in the initial iterations which are definitely not suitable for Sparse rewards scenarios. Hence, it has been seen in literature [A,B] that PPO and TRPO fail to perform optimally under sparse reward scenarios as also can be seen from our experiments.
>
> [A] A. S. Bedi, A. Parayil, J. Zhang, M. Wang, and A. Koppel. On the sample complexity and metastability of heavy-tailed policy search in continuous control. arXiv preprint349
> arXiv:2106.08414, 2021
>
> [B] S. Chakraborty, A. S. Bedi, A. Koppel, P. Tokekar, and D. Manocha. Dealing with sparse
> rewards in continuous control robotics via heavy-tailed policies. arXiv, 2022.
>
> >**Question 1:**  what is gradient tracking? it would be better to briefly summarize how [42] do this.
>
> **Reply to Question 1:** Gradient tacking is a popular technique to reduce the variance of the stochastic gradient estimates. In [42] (cf. Eq. (14)), the authors utilized a momentum based gradient tracking scheme, which requires access to two gradient evaluations at each episode. In contrast, in this work, we propose to utilize an Adam style update  [A] for gradient tracking along with heavy trailed policy parametrization and show its benefits via simulations.
>
> [A] Kingma, Diederik P., and Jimmy Ba. "Adam: A method for stochastic optimization." arXiv preprint arXiv:1412.6980 (2014).
>
> >**Question 2:** "training instability" is used a lot in this paper but never formally defined.
>
>
> **Reply to Question 2:** This is a good point. This is actually related to the sensitivity of the reward returns with respect to the hyperparameter selection for different algorithms. We think this is mostly related to the stochastic nature of the policy gradients used, which increases more for Gaussian policy parametrization with high variance and Cauchy policy in general. But we realized that we can stabilize the performance with the help of gradient tracking because it reduces the variance of the stochastic gradients. This is not a mathematical term that we want to define rigorously.

---

### Meta-Review · Area_Chair_2mEi · 2022-08-15

**Recommendation:** Accept (Poster)
**Confidence:** 4

**Metareview:**

Summary:
The paper investigates an application of heavy-tailed Deep RL to outdoor navigation tasks.  Three problems (goal reaching, obstacle avoidance, urban navigation) are tackled.  The main thrust of the paper is that heavy-tailed distributions can deal with sparse rewards better than standard distributions, and are more likely to sample a variety of actions to support learning.  Simulation and real world tasks are used to evaluate the approach.  The approach shows good performance in both simulation and reality, and compared to a set of selected baselines.

The paper is moderately original, and some extra experimentation may be needed.  The paper in well-written and understandable in general.  It is likely to represent an incremental change in state of the art and impact may be limited.  The approach is in scope for CoRL and contains sufficient robotics focus.

Strengths
•	The topic is relevant to the community
•	Results are encouraging, the algorithms performs well
•	The tasks selected as important and relevant
•	The paper is generally well written, carefully orgasnised, and has excellent supplementary material.

Weaknesses
•	The work is similar to some of the previous literature, and the amount of scientific novelty is unclear.  The main concern is that the contribution may be minor.
•	The approach uses only a single scan line, meaning most of the capabilities of the sensor aren’t used
•	Standard SLAM solutions can solve the problems without the use of learning.
•	There is no discussion on the differences between simulated and real test scenarios and how it affects algorithmic performance
•	The work should be placed into the broader context of outdoor navigation literature so the approach can be properly assessed.  This would be best done during literature review, but also by including non-learning algorithms to the baseline comparisons
•	The experimental section could be bolstered by investigating the replacement of heavy-tailed policies in other RL algorithms, in addition to adding ‘classical’ baselines as above.


***Update***
Overall, the authors have been responsive to the reviewers comments and the paper is much stronger as a result.  The reviewers are split between reject and accept, in this case I think there is sufficient interest to the research community to accept as a poster, but the area of research as a whole is worthy of much fuller studies in the future.

---

> ### Author Response · Authors · 2022-08-21
> **General Response**
>
> We thank all reviewers for their constructive feedback and insightful questions. We are particularly encouraged that they consider the proposed method is relevant and interesting. We address individual questions of reviewers in separate responses. We have also uploaded a revised version of our paper based on reviewers' suggestions. The major changes are highlighted in blue color. In particular, we have added the following modifications to the paper,
>
> 1. Training results for PPO and TRPO (as requested by reviewer **cxNk** and **B9Ro**) algorithms using heavy-tailed policies to further highlight the importance of our HTRON formulation (see Fig. 2 in the updated version of the paper uploaded with the comment).
>
> 2. We also performed additional experiments on PPO, TRPO, and Reinforce algorithms by increasing the standard deviation of their Gaussian policy distribution.  We observe that even with increasing variance, our algorithm (HTRON) outperforms all the baselines including PPO, TRPO, Reinforce, etc which demonstrates the superiority of our algorithm. Additionally, We observe that the Gaussian policies cannot achieve faster or better policy convergence by simply increasing the standard deviation of the policy distribution. Instead, the Gaussian policies demonstrate higher instability during training (see Fig. 3 in the updated version of the paper uploaded with the comment).
>
> 3. We incorporate two non-learning/classical navigation algorithms: Dynamic Window Approach(DWA) and Ego-graph to compare the navigation performance of our algorithm. While agreeing with the fact that several classical algorithms can perform successful goal-reaching and obstacle avoidance tasks, we observe that our method can achieve comparable or better navigation performance even during uneven terrain navigation (see Fig. 4,5, and Table 1).
>
> We greatly appreciate all reviewers' time and effort in reviewing our paper. We hope that our updated version of the paper and responses have addressed all reviewers' questions and concerns. Please let us know if there are further questions.

---

> > ### Author Response · Authors · 2022-08-21
> > **Updated paper**
> >
> > Please find the attachment.